# Insight into Potential Biomedical Application of Mesoporous Materials

**DOI:** 10.3390/pharmaceutics14112382

**Published:** 2022-11-04

**Authors:** Irene Alexandra Spiridon, Irina Draga Căruntu, Iuliana Spiridon, Radu Brăescu

**Affiliations:** 1Department of Morpho-Functional Sciences I-Pathology 1, “Grigore T. Popa” University of Medicine and Pharmacy, 700115 Iaşi, Romania; 2Department of Morphofunctional Sciences I-Histology, “Grigore T. Popa” University of Medicine and Pharmacy, 700115 Iaşi, Romania; 3“Petru Poni” Institute of Macromolecular Chemistry, Aleea Gr. Ghica Vodă 41A, 700487 Iaşi, Romania; 4Department of Implantology, Removable Prosthesis, Denture Technology, “Grigore T. Popa” University of Medicine and Pharmacy, 700115 Iaşi, Romania

**Keywords:** mesoporous materials, biomedical applications, drug delivery, imaging

## Abstract

The physicochemical properties of many drugs have a decisive impact on their bioavailability, as well as the pharmacokinetic efficiency in various disease therapeutics. That is why mesoporous materials have attracted a special interest in the drug delivery field, facilitating the loading of drugs into their pores due to their high surface area and porosity. The interfacial interactions established with drug molecules represent the driving force for efficient drug loading and controlled release kinetics. Moreover, these materials offer an optimal design for implantable local-delivery devices or for improving the accuracy of imaging techniques in clinical diagnosis. Their use is validated by improvements in therapeutic outcome and prevention of side effects. This review discusses the role of mesoporous materials in different biomedical applications.

## 1. Introduction

The article summarizes the recent advancements for mesoporous materials in translational science, focusing on the top two applications with direct impact on patient management: as a drug carrier for treating multiple diseases and as additional tool for improving the accuracy of imaging techniques in clinical diagnosis. The structural and chemical properties of mesoporous materials are described synthetically, providing key details without mentioning distinctive morphological features, the paper focusing on the latest trends and applications in the biomedical field.

The term porous material is used for all materials that are full of pores, holes, channels, vessels or cavities characterized by a larger depth to width ratio [1].

According to the International Union of Pure and Applied Chemistry (IUPAC), mesoporous materials are materials possessing pore dimensions in the range of 2–50 nm, which can either form a disordered or ordered network [2]. Materials with pore sizes between 2 and 50 nm are called mesoporous while materials with pores sizes smaller than 2 nm are called microporous, while nanoporous materials have pore diameters between 1 and 100 nm [3]. The large accessible surface areas and tunable pore sizes are key features of particular benefit for mass transport and dispersion of electrons/reactants [2]. The main types of mesoporous materials are: mesoporous silica; mesoporous carbon; mesoporous hydroxyapatite; hydrogel; metallic nanoparticles. A recent review [4] analyses their potential as cargo holders for different classes of therapeutic drugs with anticancer, antirheumatic or antihypertensive effect and highlights the delivery applications related to their large accessible surface areas, as well as to the tunable pore sizes.

## 2. Mesoporous Materials

### 2.1. Mesoporous Silica Materials

#### 2.1.1. Synthesis and Properties

The first synthesis method for mesoporous silica materials was reported in 1968 (Process for producing silica in the form of hollow spheres” Application No. US 342,525 A filed on 4 February 1964; Publication No. US 3,383,172 A published on 14 May 1968) [2]. Over time, the interest in this kind of materials increased and new methods of obtaining them were developed. In order to synthesize silica mesoporous materials, long chain organic surfactants are typically used as templates. The employed surfactants are anionic, cationic, non-ionic (neutral) and zwitterionic compounds, depending on the polarity of the head group [5,6]. Surfactant molecules undergo an assembly process in the presence of a silica precursor—they function as directing agents to the build-up of porous architectures in the final products [7].

The soft-templating method creates mesoporous materials with uniform morphologies and hierarchical porosity [8]. These are the result of intermolecular interaction forces that lead to template formation [9]. The inorganic materials are deposited on the surface and interior of these templates and mesoporous materials are thus obtained [8]. Another approach is to use organic surfactants and/or block copolymers that are able to interact with metal ions [10]. Thus, mesoporous materials are synthesized via sol-gel process [4,11]. The driving forces of assembling are non-covalent bonds such as hydrogen bonds, van der Waals forces and electrostatic interaction [12].

As compared to soft templating method, the hard templating method is more frequently used due to the sizes and structures of materials that are less flexible [11]. A scheme of obtaining mesoporous materials is presented in Figure 1.

High chemical stability, particle size, pore size, charge, shape, surface functionality as well as biocompatibility of these mesoporous materials make them promising carriers for targeted delivery of various encapsulated drugs [13]. These mechanical properties generally translate towards improved solubility and permeability of loaded molecules, superior bioavailability and a decreased in the side effects commonly associated with traditional drug delivery methods [13,14]. Moreover, it is worth mentioning that the material design technique can facilitate controlled release kinetics [7]. A brief overview on the synthesis of mesoporous silica nanoparticles and their use as drug delivery systems was published by Manzano and Regi [15].

The therapeutic agent is introduced inside the mesopores of the material and the pore entrances are closed with a cap once the cargo is loaded inside the cavities [7]. The cap agents are detached from the pore entrances by the action of certain stimuli in order to release the drug [16]. A schematic representation of the mechanism of therapeutic drugs release from mesoporous silica nanoparticles is presented in the Figure 2.

While the research direction of MSNs is actively investigated, with novel scaffold properties and many drugs used for loading and decoration, none of the products have been approved for medical applications yet, in spite of many studies addressing key concerns such as their degradation and biodistribution [17]. Dissolution of silica results from nucleophilic degradation by water of the siloxane and silanol groups, which produces silicic acid, biocompatible and excreted through urine [18,19]. Recent reviews focused on this topic bring structure to the previously mentioned points [20,21].

#### 2.1.2. Mesporous Silica Materials and the Blood–Brain Barrier

Avoiding the BBB can be achieved through alternate means of administration which involve exposure of brain matter through surgical procedures or through pump/catheter administration in the target area—these methods have shown limited efficiency [22]. Recent studies on nanoparticle (NP) drug delivery systems attained improved release kinetics, chemical and physical stability when silica mesoporous materials were compared to conventional encapsulated drugs [23] and associate less peri-administration risk when compared to delivery techniques such as intracerebroventricular, intracerebral or intrathecal administration [24]. This is achieved through receptor mediated transport, by binding to target proteins on the blood–brain barrier (BBB) and internalization of endothelial cells—the so-called transcellular pathway for lipophilic agents [14,25]. On the other hand, the paracellular pathways allows for transportation of hydrophilic molecules [25].

Upon administration into bloodstream, functionalized NPs adsorb apolipoprotein E or A1 [26] and polysorbate-80 (PS) coating provides an advantage by mimicking the biological actions of low-density lipoproteins [27]. The size of the particles below 100 nm represents a key feature, allowing for receptor-mediated endocytosis at the level of the capillary endothelial cell and release into brain parenchyma, the coupled therapeutic agent thus being able to penetrate the BBB and reach its target [28,29].

Calcium-dopped nanocarriers own the property of masquerading of the transporter, resulting in a decreased uptake by the reticuloendothelial system and prolonged levels of the drug in the bloodstream—creating a steadier concentration gradient that provides increased chances of permeating the BBB [30].

NPs with positive charge cross the BBB through adsorptive-mediated transcytosis [31], governed by electrostatic interactions between the cationic groups belonging to the ligands on the NPs’ surface and the negative charges of the luminal pole of the endothelial cells [31,32]. Finally, NPs conjugated with molecules that have a corresponding receptor (glucose, manitose, gluthatione, AAs) undergo carrier-mediated transport [33].

A recent study on Alzheimer disease managed to surpass the limited bioavailability of rivastigmine (RV), a drug commonly used in the treatment of this form of dementia, through loading onto calcium-doped mesoporous silica nanoparticles (MSN) functionalized with PS as a targeting ligand in a rat model [34]. The PS coating achieved a complete barrier around the nanostructures, ensuring a slow release of the therapeutic compound and ensuring a concentration of 1.88× when compared to free drug administration, while also correlating with negligible haemolytic activity [34].

Mesoporous nano-selenium release delivery system with loaded with resveratrol inhibits the aggregation of amyloid proteins and modulated oxidative stress, suppressing tau hyperphosphorylation [35]. The same paper reports that the carrier system achieved ×10 times plasma concentration levels of rivastigmine than administration alone of the drug alone and memory impairment in this mouse model was significantly alleviated by addition of resveratrol to the carrier [35]. A schematic representation of blood–brain barrier (BBB) structure and possible mechanisms of transcytosis in the case of mesoporous materials is presented in Figure 3.

#### 2.1.3. Mesoporous Silica Materials in Imaging

Non-invasive imaging techniques offer the advantage of minimal risks for the patient, while also providing a detailed characterization of the targeted lesion and its relation to surrounding tissue [36]. Most of the more detailed techniques used today rely on a contrast agent to overcome the natively lower sensitivity of the method and increase the level of detail obtained through scanning [37]. As magnetic resonance imaging (MRI) techniques are one of the best candidates for application of mesoporous material use, MSN based contrast agents have been developed in recent years in order to enhance resolution [36,38,39].

Novel methods to prepare multi-functional hollow mesoporous particles of silica without using a template have been developed. Ma et al. [40] have obtained materials with good colloidal stability, with particle sizes of about 82 nm and high surface areas by encapsulation of isopentyl acetate and superpara-magnetic nano-size particles of iron oxides. The distribution of Fe_3_O_4_ nano-size particles on the shell of the organosilicate material produces excellent imaging contrast for magnetic resonance imaging. Another study reported the potential of magnetic silica mesoporous nanoparticles as a theranostic tool for breast cancer treatment with a hydrophobic drug named exemestane [41].

In cancer related pathology, MSNs achieve satisfactory concentration within the target area due to tumour enhanced vasculature and retention effect, while also being rapidly excreted by the kidney [21,42]. Thus, loading the mesoporous scaffolds with paramagnetic complexes, an endeavour started by Taylor et al., with the development of Gd-Si-DTTA system achieved superior resolution [43], and was further developed in later years by many groups by adjusting the chemical and morphological properties of MSNs [39,44]. However, recent findings point to the possibility of integrating of nanoparticles within the MSN meshwork, generating the so called core–shell hybrids, which display a magnetically active core and are good candidates for in vivo testing [45]. As alternative to gadolinium, manganese oxides [46,47], superparamagnetic iron oxide nanoparticles (SPION) [48], Dy-DOTA [49] and binuclear ^1^H and ^19^F spectroscopy [50,51]. In the case of SPION/MSN platforms, an increase in specificity for tumour tissue has been obtained by functionalization with peptides that atherosclerotic plaques, with promising results that prove their utility outside tumour-related pathology [52].

#### 2.1.4. Mesoporous Silica Materials in Bone Pathology

Since the first application of mesoporous silicas nanoparticles in 2003, when Lai et al., used a silica scaffold to deliver vancomycin and adenosine triphosphate, these materials have proved their superiority over standard drug delivery methods [53], opening a pathway in translational sciences domains with applications in the medical field. Thus, the number of MSNs based compounds have increased considerably due to modulation of structure, composition and morphological features, and the spectrum of attached molecules is very large [38].

In the field of regenerative medicine, MSNs are one of the key development directions, particularly in bone diseases. Due to target delivery of a wide range of products used in both treatment of bacterial infection, preventing persistent biofilm formation and modulation of associated inflammation, MSNs can be loaded with compounds that promote bone tissue regeneration through cell growth and differentiation, while also guiding collagen production to ensure optimal wound healing [54]. Supplementary, bone fragility associated with conditions such as osteoporosis can be treated with MSNs, with lower drug doses that minimize the side-effects commonly encountered during parenteral or oral administration [55]. Since this condition is associated with a large disease burden, novel strategies that focus on loading different agents with synergistic effect can result in remission; this is achieved by coating the nanoparticle surface with agents such as poly (ethylenimine) (PEI) which serve as RNA carriers, while loading the pores with more traditional compounds such as osteostatin or alendronate [56,57]. A recent study on ROS-scavenger nanoceria shows promising results on pre-osteoblast cell cultures, its loading on MSNs inducing an osteogenic response in the absence of additional osteogenic supplements [58].

In oral cavity pathology, MSN type SBA-15 (Santa Barbara Amorphous) loaded with recombinant human bone morphogenetic protein-2 (rh-BMP-2) proved good hemocompatibility, with no cytotoxicity and a positive effect on human periodontal ligament cells, recorded on the basis of cell viability upregulation [59]. MSNs also associate with good penetrability and sustained antibacterial effect when loaded with clorhexidine [60,61], with promising prospects for restorative dentistry by incorporation into dental resin composites [62,63]. The translation of these observations in implantology has generated great interest in designing novel uses for mesoporous silica compounds that prevent inflammation, thus lowering the risk of complications [64]. One recent study demonstrated that mesoporous integrated titanium implants do not negatively impact the osseointegration process, and are a promising solution to prevent peri-implantitis [65]. Mesoporous silica coating of titanium implants with size-tunable pores show high stability, but also good degradability in time of the nano-interface, while adding cargo such as minocycline hydrochloride to the channels provides a superior antibacterial response when compared to bare titanium [66].

Using nanoparticles in the treatment of bone cancer relies on the principle of accumulation due to enhanced vasculature in the tumour area [67], with neoangiogenesis based on capillaries with large fenestrations that allow passage of the particles in the interstitial compartment and retention in malignant tissue [68]. However, metastatic tissue present in the bone harbours significantly reduces neoagiogenesis compared to primary tumours, making it difficult to take advantage of the vascular network for optimal drug delivery—in overcoming this fault, MSNs research has translated into the smart cargo-delivery era [69]. The materials can be homed to bony tissue by coating with bisphosphonates and loading with anti-cancer drugs [70,71]. In this sense, advances have been registered in the treatment of metastatic breast cancer [35], but promising result have also been achieved in a sandwich system using external coating of DNA aptamers and small interfering RNA and pore loading of the silica skeleton with conventional chemotherapy drug Doxorubicin [51]. To increase the specificity in eliminating cancer cells, the carriers can be engineered so that they preferentially recognize malignant targets over healthy cells [68]. This strategy can be implemented by loading MSNs with antibodies targeting overexpressed proteins, peptides or small molecules [14], that perform a homing function for the nanoparticles [72,73].

In terms of primary tumours of the bone, variants of MSNs functionalized with various surface groups react differently in regards to chemotherapy drug-release efficiency [74]. The cytotoxic effects of conventional drugs on cancer cells can be amplified by modulation of the carrier-release dynamics by pH-responsiveness, achieved through lectin conjugation [75] or enhancement with bismuth sulphide nanoparticles [76]. To increase their affinity for malignant cells, MSN carriers can also be coated with other proteins that favour their uptake, such as transferrin, since target cells overexpress the corresponding receptor [77].

Despite the multitude of potential applications, few mesoporous based compounds have progressed beyond animal in vivo studies and into clinical trial phase [78]. Among these, the best promise arises from hybrid materials such as silica-gold nanoparticles (Aurolase and Auroshell) employed in the therapy of head and neck and prostate cancers [79], but also Cornell dots used in the diagnosis of malignant tumours such as melanoma [80].

#### 2.1.5. Other Uses

Chitosan-mesoporous silica nanoparticles scaffold loaded with ciprofloxacin presents potential for bone regeneration by controlled drug release used in activation of osteoblast cells [81].

The haemostatic effect of some mesoporous silica materials was tested by in vitro blood clotting tests, as well as on the mouse tail truncation haemorrhage model [82]. Composites of mesoporous silica cotton have proved good haemostatic performance, flexible structure, strong adsorption, biological compatibility and low cost, as compared to conventional haemostatic materials [82,83].

While the forementioned applications represent areas of interest and hot-topic subdomains, the span of utility of MSN and mesoporous silica materials is much wider.

The applications of mesoporous silica particles are given in Figure 4.

### 2.2. Mesoporous Carbon Materials

#### 2.2.1. Synthesis and Properties

Mesoporous carbon materials can be obtained by soft template and hard template-based methods [84]. Different carbon sources and silica, metal oxide or polymer beads as base are used in hard template synthesis method [85]. The soft template method relies on surfactant micelles as porogenic agents and a polymerizable precursor as the carbon source [86]. When compared to hard templates that require a post-carbonization treatment, the soft template method is superior by providing ease of removing of the micelles through thermal decomposition during carbonization [87].

In the latest years, a mechanochemically-assisted templating approach was proposed in order to reduce the waste generated by soft-templating methods [88].

The synthesis of carbon materials with mesoporous structure using spherical solid gel as the template was firstly reported by Knox and co-workers [89], while Ma et al., synthesized porous carbon materials using zeolites as the template materials [90].

Due to their use for gas separation and carbon dioxide storage [91], the large-scale synthesis of carbon materials from fruit shells, coal, wood-based products [92] is beneficial. These materials are characterized by weak structural integrity, poor conductivity and mass transport, owing to the presence of heteroatoms, limitations which have been surpassed through valorisation of mesopores for different applications [84].

Mesoporous carbon with tuned structure and potential for biomedical application can be obtained from the acid treatment of sucrose using a silica template [93]. Among their advantages are high surface area, superior capacity for loading of aromatic compounds and ability to release their cardo in a pH-responsive manner according to novel studies [94].

#### 2.2.2. Mesoporous Carbon Materials as Drug Delivery

The main application in the biomedical field for mesoporous carbon is in drug-delivery solutions [95], where fine tuning of the material and release kinetics concentrate on providing an alternative to oral route administration by achieving superior bioavailability and clearance [84]. This aspect is particularly important in anticancer treatment [96]—a recent study where mesoporous carbon was loaded with celecoxib reported an improved inhibitory effect on breast cancer cell line MDA-MB-231 [97]. Moreover, in the case of breast cancer cell cultures, carbon nanospheres loaded with doxorubicin underwent endocytosis by MCF-7 cells, achieving high intracellular concentrations and leading to cell destruction and changes in morphology of tumour cells as compared to simple administration of doxorubicin [98]. Furthermore, carbon mesoporous can be light-activated, engineered to absorb radiation and transmit a photoacoustic signal useful in imaging [99]. Complex mesoporous carbon nanoplatforms using ZnO quantum dots have been designed not only to efficiently deliver chemotherapeutic agent doxorubicin, but also to modulate drug release via chemo- and photothermal stimulation, achieving high apoptosis rated in cancer cell lines [23].

In the case of ovarian cancer, in vitro studies have reported the use of oxidized mesoporous carbon nanospheres to effectively target malignant cells and tumour spheroids through dual thermo- and chemotherapy [36]. This combination of drug delivery and photothermal therapy has been effective in targeting cancer cells both in vitro and in vivo [100].

Different mesoporous carbon materials such as folic acid-conjugated magnetic mesoporous carbon material [101] or arginine-glycine-aspartic acid (cRGD) peptide-conjugated hollow mesoporous carbon [102] have proved to be efficient delivery vehicles of doxorubicin with targeting effect on cancer cells.

In terms of sustained-release drug delivery systems, nanocomposites based on the carbon/lipid duo have been engineered to deliver nimodipine, with in vivo tests demonstrating enhanced bioavailability of the drug in spite of poor solubility in water [103].

#### 2.2.3. Mesoporous Carbon Materials in Imaging

Mesoporous carbon nanomaterials can also be integrated in the concept of theranostics via their biocompatibility and physical properties such as light absorption in UV and near-IR spectrum [104]. Their optimal structure allows for stacking of fluorescent substances, a feature successfully applied ion the detection of cancer cells via aptasensors, useful in fundamental research [95]. Resolution can go as high as targeting mitochondria by using a hybrid system of magnetic mesoporous silica and carbon quantum dots, with minimum levels of toxicity for cells in various cell lines [105]. This combination can also effectively serve as drug carrier in the case of doxorubicin [106].

Thanks to initial studies showing good results, with high specificity in the detection of solid tumours and isolated cancer cells [107], recent years have seen good development in this area, with studies using carbon nanoparticle suspensions with various pigments being able to effectively identify lymph node drainage in gastric [108] and gynaecological cancers, with some materials used in clinical practice [104].

However, MRI imaging studies have generated more interest, since it a common noninvasive method to identify a tumour and its relation to the surrounding tissue [95].

Both graphene and carbon nanotubes are materials which display charge carrier mobility, used as platform for other nanoparticles in order to increase the sensitivity [109]. Thus, mesoporous carbon nanomaterials have distinguished themselves as carriers [110] for manganese oxides [109], gadolinium [111] and iron oxides [13], with improved tumour resolution. Several reviews tackle these topics in general or focusing on a particular type of malignancy [104,112].

The general research direction in the case of mesoporous carbon seems to be oriented in generating nanoplatform and hybrid compounds which serve the classical drug carrier function, but also increase efficacy through other therapeutic or imaging methods.

### 2.3. Hydrogels

#### 2.3.1. Synthesis and Properties

Hydrogels are three-dimensional structures consisting of cross-linked hydrophilic polymer chain networks. Hydrogels distinguish themselves among mesoporous materials due to their increased water content and soft consistency, which simulates natural tissue perhaps more than any other class of biomaterials [113]. Their properties are influenced by the nature of the cross-linkage, as well as by the behaviour of the constituent polymers. The many manufacturing techniques can yield chemically stable materials or hydrogels with tunable degradability, making them excellent candidates for biomedical applications [114].

Hydrogels presents different crosslink density, which influences the network porosity and their morphology. These materials present swelling behaviour when introduced to water or physiological fluids. Their network can also be loaded with microstructures and nanoparticles which complement the mechanical properties of the hydrogel, their release kinetics making them ideal cargo holders [115].

Hydrogels are able to revert to their initial state when removed from the presence of water/biological fluids. This unique property makes elicits great interest for their use in biomedical applications such as drug delivery, drug release and vaccine design. In order to have medical applications these materials need to meet a strict set of criteria presented in Figure 5.

Moreover, their administration is not limited to the oral route, but can be extended to rectal administration in the form of suppositories, through vaginal route, in the form of tablets, ocular as drops and circular inserts [116]; by far, the most convenient application of hydrogels is in the form of wound dressing [117]. Many hydrogel products have received approval of superior forums in terms of safety and are commercially available [118]. The following section will focus on the classical directions of valorising hydrogels and updates in the field.

Hydrogels can be classified according to presence of electric charge, crosslinked junctions and type of crosslinking. Thus, in relation to the electric charges located on the chains, hydrogels are subdivided into ionic hydrogels, non-ionic hydrogels and ampholytic hydrogels [119]. When cross-linked junctions are considered, hydrogels are classified into blend hydrogels, block co-polymeric hydrogels and polyelectrolyte complex hydrogels [120]. In regards to the kind of cross-linking, hydrogels could be physically crosslinked or chemically crosslinked.

Hydrogels physically crosslinked are formed by noncovalent interactions such as electrostatic interaction, hydrogen bonding and hydrophobic interactions [119].

Chemically crosslinked hydrogels can be synthesized by use of chemical crosslinkers, that promote gelation during the polymerization of precursor building blocks. It is worth to mention that due to their low impact on the environment, physically crosslinked hydrogels are more interesting for potential applications as compared to chemically crosslinked hydrogels [121].

An element of novelty, hydrogels comprising metal nanoparticles have been designed [122]. The addition of nanoparticles into hydrogels yields new platforms that can be used for selective release under certain stimuli, being of high interest for the medical community by offering solutions addressing different human diseases. These platforms are sensitive and certain triggers are used to degrade the gel, leading to drug release [123]. It has been reported that the drug release is influenced by the variation of crosslinker substituents, use of homogenous polymer networks, as well as by the control of polymer cleavage rates [122,124].

Hydrogels facilitate self-healing process due to their ability to sequester water molecules without self-dissolving. Moreover, interactions between polymer chains and metal ions can assure the assembly forces that promote self-healing [125,126]. The ability of polymeric chains to form multiple bond interactions in and around the damaged area will allow the formation of new connections between the components that restore the material integrity [127]. A schematic illustration of hydrogels applications is presented in Figure 6.

#### 2.3.2. Hydrogels as Wound Dressing

An excellent review related to novel hydrogels wound dressing [128] evidenced their future potential in the wound care sector. Currently, there are numerous products commercially available which are used in the treatment of low suppuration wounds, ulcers associated with diabetes or increased pressure, but also skin damage in the context of abrasion wounds [129]. All of them ensure the necessary moist environment for wound healing and enhancing debridement through autolysis, thus promoting epithelization and granulation tissue formation [130]. Due to their variable degree of hydration, hydrogels have been employed in wound dressing, where absorption of excess exudate and byproducts of wound healing is complemented by mechanical protection from external factors [131]. Another important issue to be address by modern materials that replace the classical dressing is to ensure the lowest number of changes and facilitate removal once the healing process has subsided [132]. In this form, hydrogels ensure moisture in the wound environment, an important factor in the healing process [133], while also permitting loading with various substances such as antibiotics or other molecules essential to tissue regeneration [134]. The release of these drugs is subject to various mechanisms: diffusion, swelling, chemical control or environmental responsive release [135]. Moreover, the hydrogel compositions can be employed as active, iontophoretic delivery systems or passive transdermal reservoirs [115].

Advances in the technical aspect focus on loading particles that can combat infection and modulate inflammation, while also preventing biofilm formation [136]. In recent years, dressings with a controlled silver particle release have been commercially available [137], but efforts are being directed towards increasing the longevity of the dressing in the treatment of burns by hybrids [138]. Antibiotic resistance has prompted new solutions, with the development of antimicrobial peptide [139] loaded onto hydrogels that prove efficient against numerous pathogens, including methicillin-resistant *S. aureus* (MRSA) and multidrug-resistant *Escherichia coli*, while also maintaining active levels in peripheral blood of up to 48 h [140,141]. In a recent report, chronic wound healing can benefit from the addition of natural extracts of *Artemisia argyi* which modulate chronic inflammation, a process responsible for fibrosis and its negative implications [142]. Using extrusion cryogenic 3D printing technology, a new hydrogel that accelerated diabetic wound healing through promotion of angiogenesis was obtained [143].

#### 2.3.3. Hydrogels as Contact Lenses

Another fitting application for hydrogels is represented by contact lenses. Ocular disease is becoming more prevalent in the general population, especially in what chronic/age related changes such as cataract, glaucoma and macular degeneration [144]. Most ophthalmologic afflictions are treated with synchronous topical administration of various drugs, which poses certain disadvantages due to low permeability of corneal tissue, tear turnover and other factors leading to overall decreased bioavailability of the administered compounds [145]. Contact lenses that are conventionally soaked in drugs generate a burst-type release that does not maintain a steady concentration [146]; thus, the interest in the biotechnology of the hydrogel that forms the lens has led to novel discoveries in the field, synthetized in the review by Peral et al. [147]. Among the newest, drug-loaded nanoparticles incorporation of both hydrophobic and hydrophilic substances into hydrogels and ensure a longer release, while also avoiding degradation through ocular enzymes [148]. For the treatment of glaucoma, animal studies reveal satisfactory intraocular concentration of compounds achieved through a propoxylated glyceryl triacylate (PGT) contact lens with incorporated timolol [149], but also by incorporation of a latanoprost loaded on poly (lactic-co-glycolic) acid (PLGA) nanoparticles. Xu et al., have prepared PEG-PLA micellle-laden lense with timolol and latanoprost, achieving satisfactory levels still detectable four days after initial administration [148].

#### 2.3.4. Hydrogels for Bone Regeneration

Glass nanoparticles were used to reinforce the biological and mechanical characteristics of alginate dialdehyde-gelatin hydrogel. It was proved that glass particles served as carrier for icariin. This 3D printed hydrogel enhanced osteoblast proliferation, adhesion and differentiation [150]. An injectable mesoporous bioactive glass/fibrin glue hydrogel [151] was tested for osteogenic properties. The reported results evidenced that the injection of 1% composite hydrogel has promoted the formation of bone, mineralization and angiogenesis performance, but also inhibited osteoclastogenesis. Hydroxyapatite nanoparticles and calcium carbonate microspheres were added into a hydrogel-based polysaccharide (oxidized alginate and carboxymethyl chitosan) with tetracycline hydrochloride embedded to an end injectable gel scaffold with demonstrated antibacterial activity against *E. coli* and *S. aureus*, as well as more efficient drug release [152]. The mechanical and biological properties recommend this gel for bone regeneration applications.

#### 2.3.5. Hydrogels as Drug Delivery

A drug embedded into a carrier may address the above shortcomings to a certain extent. That is why, hydrogels able to deliver drug release at the lesion sites are beneficial due to the improvement of the curative treatment, as well as to their capacity to reduce side-effects. The release mechanisms from hydrogels depend on hydrogel composition, or solute properties. Moreover, swelling, change in pH and temperature variations influence the targeted drug release. Attempts to overcome the challenges of delivering adequate concentrations to the more profound structures of the eye have prompted the idea of intravitreous administration of hydrogels, a more recent paper listing the advantages and drawbacks of using these mesoporous materials in this area of pathology [153].

Cytarabine and methotrexate were encapsulated into a hydrogel based on mesoporous silica nanoparticles, sodium hyaluronate, chitosan and carboxymethylcellulose. The system was reported as an efficient dual-responsive dual-drug delivery platform that presents an excellent biocompatibility and exhibits a growth inhibitory effect on hepatoma (HepG2) cells [154]. When doxorubicin and bortezomid were added into a polymeric nanogel with glutathione-responsive dissociation [155], the intracellular release of drugs was accelerated, their anti-tumour therapeutic effects being enhanced. It seems that the superior drug delivery at target site is due to the destroying of nanogel structure at elevated glutathione levels. Caramella et al. [156] highlighted that mucoadhesive and thermogelling delivery systems could be beneficial for improving the efficacy of vaginal formulations.

#### 2.3.6. Hydrogels in Imaging

Rhodamin B isothiocyanate-labelled mesoporous silica nanospheres with magnetic nanoparticle core and doxorubicin were embedded into a self-healing hydrogel prepared from Schiff base bonds between pullulan and chitosan. The material can be used in the tumour diagnosis and exhibited a efficient synergistic magnetothermal-chemo-chemodynamic in the cancer therapy [157]. Doxorubicin was added into a nanogel consisting of boronic acid-conjugated lactose modified-chitosan and dopamine- and nitric oxide-conjugated partially carbonized hyaluronic acid [158]. It was found the developed material can be used in imaging, while the targeting ability against malignant tumours via pH/enzyme-responsive moieties prove its potential in the cancer monitoring.

#### 2.3.7. Other Uses

In view of current trends, however, a new direction focused on combining the advantages of mesoporous materials in order to yield the best results has given rise to hybrid materials such as composite hydrogels with nanosized silver bromide-doped mesoporous silica, accompanied by effective antimicrobial properties and improved healing of skin confirmed through histological examination [138]. The silica-hydrogel system can be effective in both therapy and diagnostics, as highlighted by a recent paper, where synthetic, fibril-forming peptides were taken up efficiently by cancer cells and released their cargo in the intracellular compartment [159]. In other study, lignin metallic and bimetallic (silver and gold) nanoconjugates were incorporated into a poly (acrylic acid) hydrogel [160] in order to improve the cell killing effect of the nanoconjugates. This study has demonstrated that obtained hydrogels presented pH triggered sustained release and exhibited greater photodynamic activity on microbial cells.

Colchicine encapsulated in mesoporous silica nanoparticles was loaded in a hydrogel obtained by reacting carboxyethyl chitosan and oxidized pullulan and then deposited onto cotton fabric in order to obtain transdermal patches. These prolonged-release formulations were useful in the treatment of osteoarthritis [161].

### 2.4. Metallic Nanoparticles

Usually, these nanoparticles are synthesized by chemical methods such as the sol-gel process, chemical precipitation method, chemical vapor deposition, hydrothermal synthesis, sonochemical method or polyol synthesis. The most used *sol-gel process* involves the hydrolysis of a metal oxide solution to produce a colloidal suspension (sol) which is transformed by condensation process into a gel [162]. Gel phase is an interconnected network of solid particles that form a continuous entity throughout a liquid phase. Nanoparticles with well-defined structures, complex shapes and a high specific surface area are obtained by drying or heating [163].

Using the *precipitation method*, the metal nanoparticles are obtained through mixing the precipitating agent into the metal-salt solution under stirring conditions, at room temperature or higher values [12].

*Chemical vapor deposition* is based on the influence of high temperatures on the precursor materials which are exposed in the gaseous state on substrates which are heated. The precursors react or decompose on the substrate surface to form nanoparticles [164].

*Hydrothermal method* facilitates the obtainment of metal nanoparticles by use of high values of temperature or pressure to dissolve and recrystallize metal precursors. There are various variants techniques, such as hydrothermal-electrochemical, hydrothermal-ultrasonic hydrothermal-microwave or hydrothermal-sol-gel technique [165]. In a closed reaction system, these methods achieve the high temperature and high pressure in shorter times as compared to hydrothermal method [166,167].

Synthesis of metal nanoparticles by use of *sonochemical methods* is based on the breaks the chemical bonds of the starting material (metallic salts), which are generated under effect of the intensified ultrasonic vibrations. As result of alternate compression and relaxation of the metal salt solution, metal nanoparticles with uniform size distribution and high surface area are obtained [167]. Ultrasound based techniques carry the advantage of reproducibility [168].

*Polyol method* has a low production cost and allows a good dimension controllability and efficiency. A cold solution containing metal salt (metal precursor) and poly (vinylpyrrolidone) as polymeric capping agent is added into ethylene glycol (polyol) at an elevated temperature. Poly (vinylpyrrolidone) is used as directing agent and also prevents the aggregation of the nanoparticles [169].

Metal nanoparticles have unique properties which recommend them for medical and bioimaging applications. Their size allows them to access the circulatory/lymphatic systems, being able to deliver different drugs or other biomolecules to various cells and tissues, particularly cancer cell [170].

#### 2.4.1. Metal Nanoparticles in Cancer Therapy

The delivery efficiency of metal nanoparticles to the target tumour site is greatly affected by several factors including tumour-specific-targeting efficiency, distribution of nanoparticles in the body, as well as the circulation time. These factors are governed by the size, shape, surface coating and charge of nanoparticles. Therefore, the smaller the gold nanorods are, the more promising they become for in vivo mesoporous applications compared to other metal nanoparticle structures. Golden nanoparticles can scatter and efficiently absorb incidental light at the resonance wavelength [171]. In this regard, mesoporous microspheres of hydroxyapatite-gold containing methotrexate [66] were designed. The materials exhibited a high therapeutic efficacy, being a promising platform for cancer therapy. We did not find FDA-approved clinical trials of gold nanoparticles for medical theranostic applications, but Libutti et al. [172] reported that citrate-coated 27 nm gold NPs bound with thiolated PEG and TNF-a presented tumour targeting and tumour toxicity dual effects in a clinical trial performed in 2005. The National Cancer Institute (NCI) has planned for phase II clinical studies for CYT-6091 (CytImmune, https://www.cytimmune.com/pipelilne, accessed on 21 July 2022).

Mesoporous porphyrinic metal-organic nanoparticles were used to develop scaffolds as a platform for bone tumour therapy. It was found that these materials are able to kill MG-63 cells in vitro and to inhibit subcutaneous tumour growth in vivo [173]. The bone-targeting ability resulting in the inhibition of cancer cells obtained by mesoporous glass coated with hyaluronic acid-alendronate was reported [174]. pH-dependency drug release from these materials is important in the treatment of bone metastasis. Bone-mimetic selenium-doped hydroxyapatite nanoparticles have induced the apoptosis of cancer cells [175] due to increased selenium-induced ROS generation in tumour tissue. Although metal mesoporous nanoparticles alone are efficient in medical treatment, they are currently regarded as synergetic with different therapeutic methods considered classical in the management of cancer, such as chemotherapy, radiotherapy or immunotherapy [176]. The metal nanoparticles can generate reactive oxygen species in intracellular compartment, thus negatively impacting the target tissue by exposure to supplementary oxidative damage [177]. This phenomenon is regarded as the potential toxic effect of these nanoparticles, its acknowledgment proving insufficient in the absence of deeper knowledge regarding the intricate interactions between metallic nanoparticles and biological systems [178].

Mesoporous silica nanoparticles-silver nanoparticles have proved their efficiency in the detection of breast cancer cells [179] reported that carbon-doped TiO_2_ nanoparticles have suppressed the proliferation of 4T1 breast cancer cell line in both in vitro and in vivo models in combination with ultrasound treatment. Doxorubicin was loaded to mesoporous platinum modified with polyethylene glycol [180]. It was found that upon laser irradiation, the anticancer effect was significantly improved, proving that this material could be used in combined chemotherapy and photothermals treatment for cancer. In another study, copper ferrite nanoparticles and oxaliplatin, modified by polydopamine via alkaline-triggered polymerization proved able to catalyse the Fenton reaction inside the tumour cells, inhibiting the malignant growth more effectively than free oxaliplatin [181]. In another study, a biomimetic nanodrug delivery system based on cancer cell membrane-coated gold nanocage was designed [176], exhibiting an unprecedented tumour-targeting ability. These novel systems represent possible solutions for the future therapy of breast cancer, warranted they require further studies.

#### 2.4.2. Metal Nanoparticles in Imaging

Tuli et al. [182] highlighted that metallic nanoparticles are able to modulate the expression of various signalling molecules in the tumour microenvironment. Magnetic nanoparticles are well recognized for their capacity to bring multifunctionality in the construction of nanotheranostics. Fe_3_O_4_@TiO_2_–ZrO_2_@mSiO_2_ have detected endogenous phosphopeptides profile in human saliva [183], opening new possibilities in different disease biomarker-based diagnosis. Mn-based theranostic nanoplatforms have received interest due to their potential for imaging and cancer therapy [184]. It is worth to mention that most in vivo experiments have been performed on mice and clinical translation is still under exploration. Hollow mesoporous manganese oxides are able to achieve good selectivity, a well-known challenge in oncological treatment, and to target the release on tumour site [185]. Moreover, it was proven that contrast agents comprising mesoporous manganese oxides materials are able to alleviate tumour hypoxia and enhance magnetic resonance imaging [170]. Polydopamine-modified Fe_3_O_4_ nanocomposites entirely eradicated the 4T1 tumour in bearing mice. Moreover, they exhibited an excellent MRI contrast effect [186], much higher as compared with commercially available MRI contrast agents. The potential of FePt/SiO2/Au hybrid nanoparticles to improve the resolution, as well as the difference between healthy and cancerous tissues were clearly demonstrated by in vitro MRI experiments [187]. A new material comprising poly (ethylene glycol)- molybdenum disulfide–gold nanoparticles loaded chlorin e6 was efficient in the release the drug around the tumour site [188]. Under heat influence, Chlorin e6 was released from the system and has intensified/augmented NIR fluorescence signals. Thus, the synergy established between the photodynamic and photothermal effect for anti-tumour therapy was achieved.

#### 2.4.3. Other Uses

Zinc oxide nanoparticles were efficient in the treatment of infectious diseases, as well as in the diabetes and cholesterol issues [189]. A recent review article mentions the role of nanoparticles as immunomodulators in the treatment of inflammation-related disorders [190]. A schematic representation of their action is given in Figure 7.

## 3. Conclusions

It is anticipated that the interest for mesoporous materials will continue to increase, as current results show great promise, especially for medical applications.

Standardization of clinical evaluation methods employing mesoporous materials nanocarriers, as well as a guide related to their properties (particle size, pore size and colloidal stability) becomes mandatory in order to have a realistic image of their contribution to targeted applications. Nevertheless, the literature data have evidence of the high potential of mesoporous materials for effective and efficient management of different diseases. Further research works are needed in order to improve biological safety and to minimize the adverse effects of mesoporous materials, thus facilitating the preclinical and clinical trials in diagnosis and treatment processes. 

## Figures and Tables

**Figure 1 pharmaceutics-14-02382-f001:**
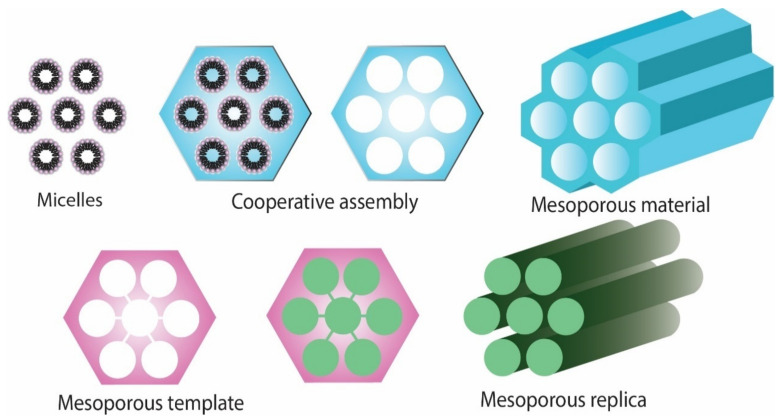
Schematic representation of obtaining mesoporous materials.

**Figure 2 pharmaceutics-14-02382-f002:**
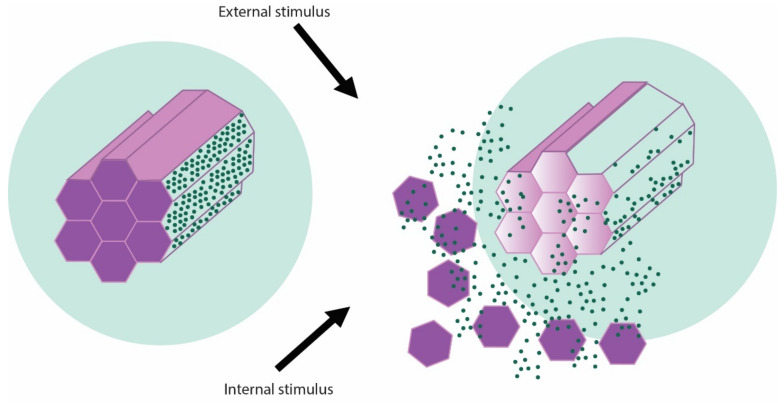
Schematic representation of stimuli-generated release of therapeutic cargo from MSNs.

**Figure 3 pharmaceutics-14-02382-f003:**
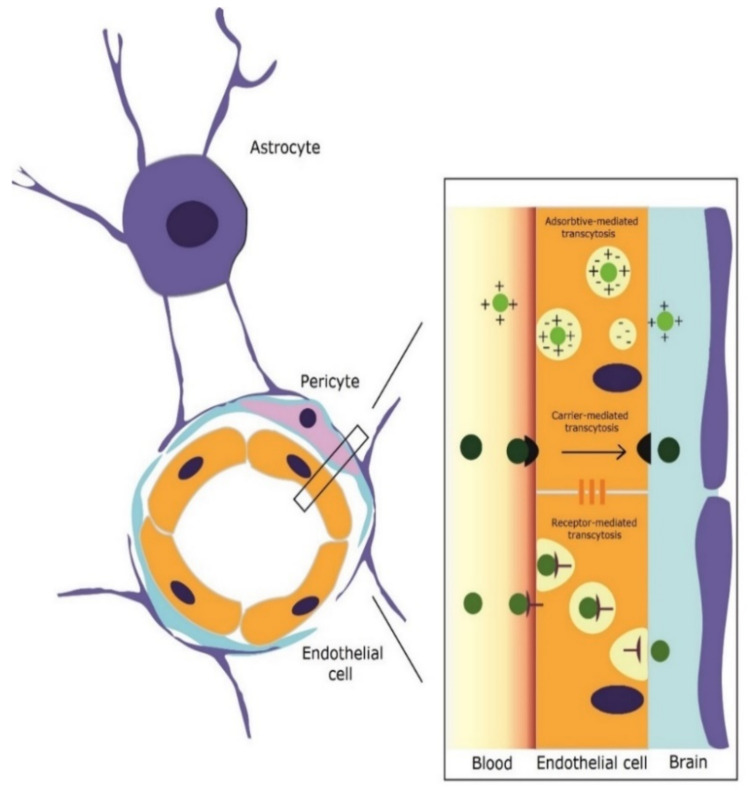
Schematic representation of blood–brain barrier (BBB) structure and possible mechanisms of transcytosis in the case of mesoporous materials.

**Figure 4 pharmaceutics-14-02382-f004:**
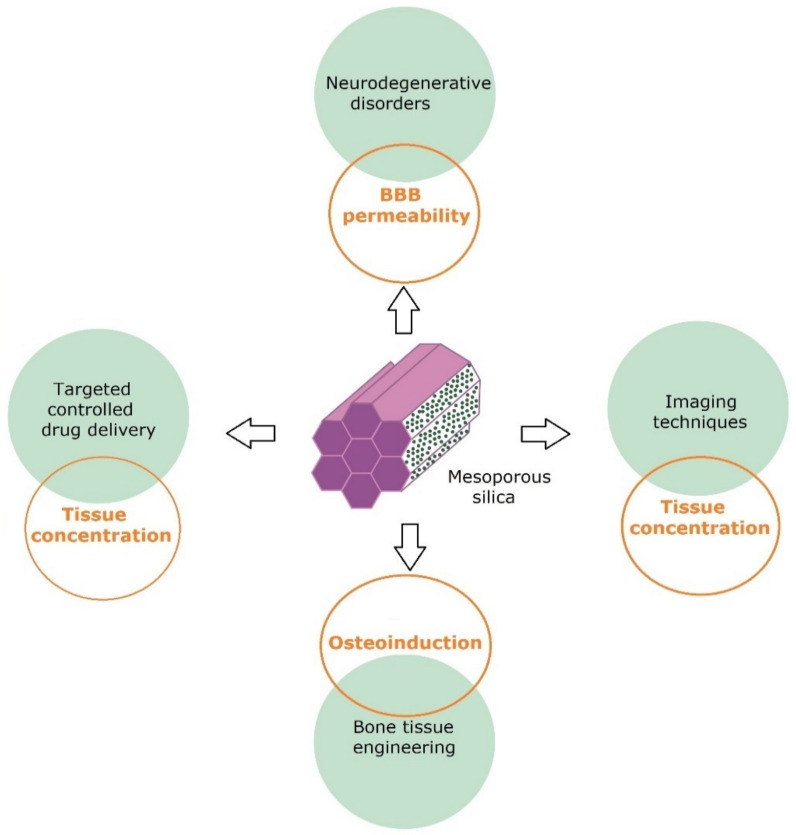
Mesoporous silica applications and main challenges in their field.

**Figure 5 pharmaceutics-14-02382-f005:**
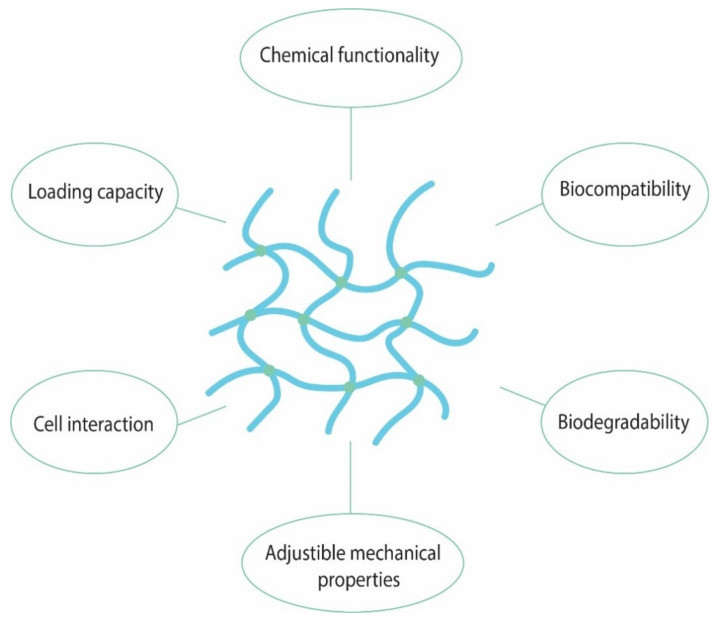
The main characteristics of hydrogels with medical applications.

**Figure 6 pharmaceutics-14-02382-f006:**
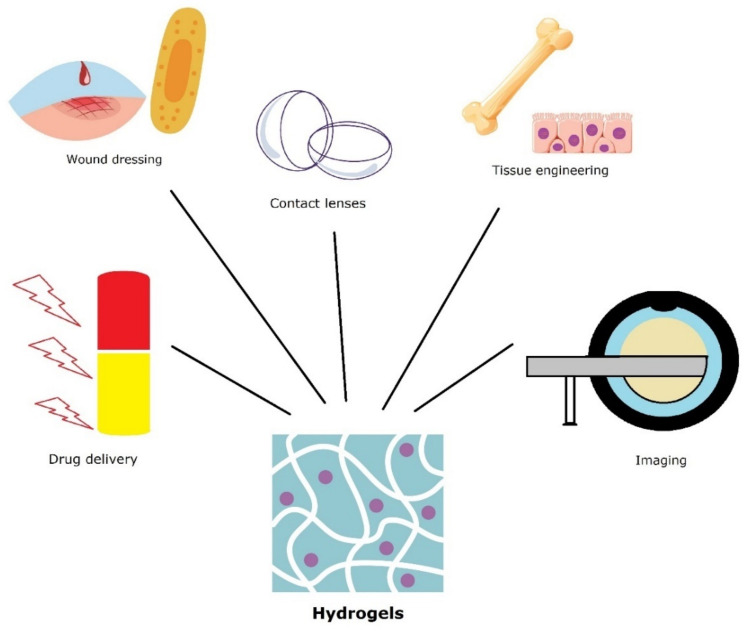
Various applications of hydrogel mesoporous materials.

**Figure 7 pharmaceutics-14-02382-f007:**
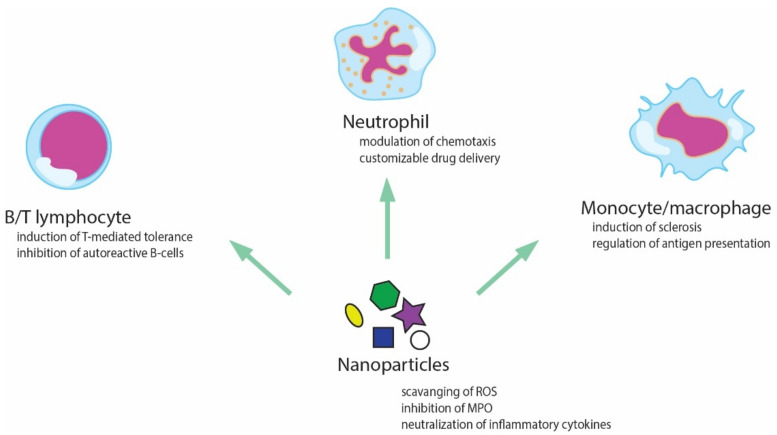
Possible interactions between nanoparticles and different components of the immune system.

## Data Availability

Not applicable.

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
