# Peer review of "Insight into Potential Biomedical Application of Mesoporous Materials"

_pharmaceutics, 2022, doi:10.3390/pharmaceutics14112382_

Round 1

Reviewer 1 Report

The present review tries to present a compilation of the recent advances in the use of mesoporous materials for drug delivery and imaging applications. The authors take a great effort to summarize most of the direct and/or potential applications of mesoporous materials, and elaborate a vast list of references.

However, this presentation of many points dilutes the intended focusing points (drug delivery and imaging). Importantly, the reduced list of Figures (just 4), none of them summary of previous works, makes this work a not so useful tool for a reader. Furthermore,  the high number of more complete recent reviews on the field yields a low significance of the contribution.

Due to this, unfortunately my decission is to reject this review for publication in Pharmaceutics.

Author Response

Rewiewer 1

The present review tries to present a compilation of the recent advances in the use of mesoporous materials for drug delivery and imaging applications. The authors take a great effort to summarize most of the direct and/or potential applications of mesoporous materials, and elaborate a vast list of references. However, this presentation of many points dilutes the intended focusing points (drug delivery and imaging). Importantly, the reduced list of Figures (just 4), none of them summary of previous works, makes this work a not so useful tool for a reader. Furthermore, the high number of more complete recent reviews on the field yields a low significance of the contribution. Due to this, unfortunately my decission is to reject this review for publication in Pharmaceutics.

Response: Thank you for the time and effort they spent reviewing our manuscript.

Reviewer 2 Report

The manuscript involves a literary review of scientific research progress used in potential biomedical application of mesoporous materials. I appreciate the efforts of the Authors and admire their work. It is well structured and has been written in a way to facilitate the understanding of the reader. I recommend publishing the manuscript without any modification.

Author Response

The manuscript involves a literary review of scientific research progress used in potential biomedical application of mesoporous materials. I appreciate the efforts of the Authors and admire their work. It is well structured and has been written in a way to facilitate the understanding of the reader. I recommend publishing the manuscript without any modification.

Response: Thank you for the time and effort they spent reviewing our manuscript.

Reviewer 3 Report

In the manuscript pharmaceutics-1933186, the authors have comprehensively reviewed the state of art progress made in the thematic domain of mesoporous materials in the direction of biomedical application. The manuscript is interesting, however, the authors need to address the following concerns: 

1. A detailed table for commonly used abbreviations should be presented. Some abbreviations are not defined in the first instance e.g. BBB. Check carefully.

2. The authors have not adequately used graphics (from the referenced papers) to illustrate the various process and results. To better understand the subject, at least 1 cluster figure should be included in each section. 

3. I have noticed some repetitions of sections. E.g. 

Section 2.2.2 and section 2.3.2

Section 2.2.3 and section 2.3.3

4. Check if hydrogels and metallic nanoparticles are really a category of ‘mesoporous materials’ which is the core theme and title of the paper.

Author Response

In the manuscript pharmaceutics-1933186, the authors have comprehensively reviewed the state of art progress made in the thematic domain of mesoporous materials in the direction of biomedical application. The manuscript is interesting, however, the authors need to address the following concerns: 

  1. A detailed table for commonly used abbreviations should be presented. Some abbreviations are not defined in the first instance e.g. BBB. Check carefully.

Response: Thank you for your observation. The whole manuscript was carefully checked and all abbreviations were explained at the end of manuscript.

  1. The authors have not adequately used graphics (from the referenced papers) to illustrate the various process and results. To better understand the subject, at least 1 cluster figure should be included in each section. 

Response: Several figures that summarize the information presented in each section have been inserted.

  1. I have noticed some repetitions of sections. E.g. 

Section 2.2.2 and section 2.3.2

Section 2.2.3 and section 2.3.3

Response: Thank you for your observation. The text was carefully checked and the whole manuscript was corrected accordingly.

  1. Check if hydrogels and metallic nanoparticles are really a category of ‘mesoporous materials’ which is the core theme and title of the paper.

Response: Usually, hydrogels are materials developed at tailored dimensions, as function of potential applications. These are a class of highly hydrated three dimensional networks made of either synthetic or natural polymers that can mimic extracellular matrix. Owing to their porosity, hydrogels are able to load different agents and release them in over a long period to reduce their toxicity and overcome the drug resistance problem.  Some traditional hydrogels suffer from low mechanical properties and low bioactivity. Their physicochemical properties can be improved by embedding metallic particles (nano- or meso- sized).  in their structure. Literature data confirm mesoporous structure of hydrogels used in medical applications.

Reviewer 4 Report

This review discusses the role of mesoporous materials in different biomedical applications. The manuscript may be accepted for publication after some minor additions/corrections.

1) Section 2.2: Though not related to bio-imaging, carbon nanotubes are also used for gas separation and CO2 storage (cite e.g., https://doi.org/10.1021/acs.jpcc.0c04325).

2) It seems Figure 1 is not cited in the text.

Author Response

This review discusses the role of mesoporous materials in different biomedical applications. The manuscript may be accepted for publication after some minor additions/corrections.

1) Section 2.2: Though not related to bio-imaging, carbon nanotubes are also used for gas separation and CO2 storage (cite e.g., https://doi.org/10.1021/acs.jpcc.0c04325).

Response: The information was inserted into the text manuscript, as follows:

Due to their use for gas separation and carbon dioxide storage [91], the large scale synthesis of carbon materials from fruit shells, coal, wood-based products [92] is beneficial. These materials are characterized by weak structural integrity, poor conductivity and mass transport, owing to the presence of heteroatoms, limitations which have been surpassed through valorization of mesopores for different applications [84].

2) It seems Figure 1 is not cited in the text.

The reference to Figure 1 has been introduced in the manuscript text, as follows: A scheme of obtaining mesoporous materials is presented in Figure 1.

Round 2

Reviewer 1 Report

The authors have throuroughly revised their manuscript. Therefore, I consider it finally acceptable for publication in the present form.